# ENHANCING EXPERIMENTAL SIGNALS IN SINGLE-CELL RNA-SEQUENCING DATA USING GRAPH SIGNAL PROCESSING

**Daniel B. Burkhardt**[1,†]**, Jay S. Stanley III**[3,†]**, Ana Luisa Perdigoto**[4]**, Scott A. Gigante**[3]**,**

**Kevan C. Herold**[4]**, Guy Wolf**[4,‡]**, Antonio J. Giraldez**[1,‡]**, David van Dijk**[1,2,‡*]**,**

**Smita Krishnaswamy**[1,2,‡*]

[1]Departments of Genetics, [2]Computer Science, [3]Computational Biology & Bioinformatics
[4]Immunobiology and Internal Medicine, Yale University, New Haven, CT, USA
[5]Department of Mathematics and Statistics, Université de Montréal, Montreal, QC, Canada

[*]Corresponding authors. E-mail: smita.krishnaswamy@yale.edu, david.vandijk@yale.edu
[†] These authors contributed equally. [‡] These authors contributed equally.

## ABSTRACT

Single-cell RNA-sequencing (scRNA-seq) is a powerful tool for analyzing biological systems. However, due to biological and technical noise, quantifying the effects of multiple experimental conditions presents an analytical challenge. To overcome this challenge, we developed MELD: Manifold Enhancement of Latent Dimensions. MELD leverages tools from graph signal processing to learn a latent dimension within the data, which scores the prototypicality of each datapoint with respect to experimental or control conditions. We call this dimension the Enhanced Experimental Signal (EES). MELD learns the EES by filtering the noisy categorical experimental label in the graph frequency domain to recover a smooth signal with continuous values. This method can be used to identify signature genes that vary between conditions and identify which cell types are most affected by a given perturbation. We demonstrate the advantages of MELD analysis in two biological datasets, including T-cell activation in response to antibody-coated beads and treatment of human pancreatic islet cells with interferon gamma.

## 1 INTRODUCTION

As single-cell RNA-sequencing (scRNA-seq) has become more accessible, design of single cell experiments has become increasingly complex. However, quantifying the differences between single cell data sets collected from different conditions presents an analytical challenge. There is often a large overlap between single-cell profiles across conditions and single cell data sets are prone to biological and technical noise. As a result, the signal of an experimental perturbation is small with respect to the biological and technical variation in an experiment (**Fig. 1**).

To quantify the differences between experimental conditions, it would be helpful to find groups of cells that are prototypical of experimental or control conditions. Thus, we effectively want a quantification (*i.e.* a score) of how prototypical each cell is of the control or experimental condition. Such a score would identify the cells and populations that are the most or least affected by an experimental perturbation. We term this score the *Enhanced Experimental Signal* (EES). For example, in a simple experiment with one experimental condition and one control condition, we would like the EES to be +1 or -1 for cells that are most likely to arise in the experimental or control condition, respectively, and 0 for cells equally likely to arise in either condition.

To derive this score, we developed MELD (Manifold Enhancement of Latent Dimensions). MELD is based on methods from graph signal processing (GSP) that, despite their proven strength in other domains, have not often been used in biomedical data analysis (Shuman et al., 2013).The key advantage of GSP is the access to a set of tools for processing *graph signals*, which are functions defined over the nodes in a graph.

MELD models the condition label indicating from which condition each cell was sampled as a graph signal that we call the *Raw Experimental Signal*. In a two-sample experiment, the RES would be defined as -1 for cells from the control condition and +1 for cells in the experimental condition. To remove high-frequency noise from the RES, MELD applies a novel filter over the graph frequency domain of the RES to infer the EES. Finally, we incorporate information from the RES and EES into Vertex Frequency Clustering (VFC), a novel clustering algorithm that identifies cell types most or least affected by the experimental perturbation.

MELD has wide applicability in the analysis of high dimensional single cell data. Here, we describe the algorithms for MELD and VFC and demonstrate the methods on two scRNA-seq datasets. More results can be found in Burkhardt et al. (2019).

## 2 THE MELD ALGORITHM

The goal of the MELD algorithm is to use a manifold model of cellular states across experimental conditions to learn an *Enhanced Experimental Signal* (EES) that quantifies how prototypical each cell is of each experimental condition.

The MELD algorithm computes the EES in the following steps:

1. A cell similarity graph is constructed over the data.
2. The experimental label that indicates the sample origin of the cell is modeled over the graph as a discrete signal called the *Raw Experimental Signal (RES)*.
3. MELD filters biological and technical noise from the RES to infer the EES, which reflects how prototypical each cell is of each condition.
4. The EES and Fourier transform of the RES are used to identify cell populations that are prototypical of each condition and to infer gene trends of the experimental perturbation.

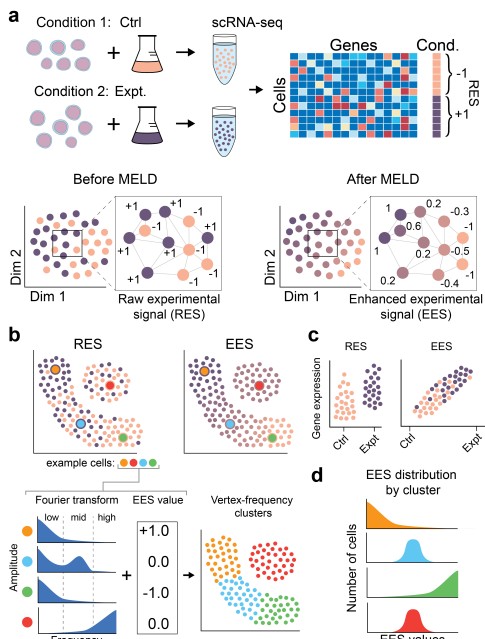

Figure 1: **(a)** MELD smooths experimental labels in scRNA-seq data. **(b)** The EES and Fourier transform of the RES are used for Vertex Frequency Clustering. **(c)** The EES can be used to infer gene trends in each cluster and **(d)** identify clusters most affected by a perturbation.

The cell similarity graph in step one is constructed using a variant of the radial basis kernel called the $\alpha$-decay kernel, first proposed by Moon et al. (2018). Next, MELD uses the input experimental label to create the RES on the graph. For simple two-sample experimental cases, cells from the control condition are assigned a value of -1 and cells from the experimental signal are assigned +1 (**Fig. 1a**). For more complex cases, such as in a time course or a series of drug titrations, the raw signal can be defined ordinally as the stage or timepoint of collection or dosage of a drug.

The goal of MELD is to remove biological and technical noise from the RES using a low-pass filter. The MELD filter is inspired by *Laplacian regularization* (Ando & Zhang, 2007), which is expressed as the optimization

$$\mathbf{y} = \underset{\mathbf{z}}{\operatorname{argmin}} \underbrace{\|\mathbf{x} - \mathbf{z}\|_2^2}_{a} + \underbrace{\beta \mathbf{z}^T \mathcal{L} \mathbf{z}}_{b}, \quad (1)$$

Here, the regularization maximizes (a) reconstruction, and (b) smoothness of **z** quantified using the graph Laplacian $\mathcal{L}$.

However, low-pass filters are not a panacea. Indeed, low frequency noise (such as background noise) is common and will be exacerbated by low-pass filtering. In MELD we propose a new class of graph filters, of which Laplacian regularization is a subfilter, that is adaptable to graph and signal noise context, given by the following equation:

$$\mathbf{y} = \operatorname*{argmin}_{\mathbf{z}} \|x - \mathbf{z}\|_2^2 + \mathbf{z}^T \mathcal{L}_* \mathbf{z} \qquad (2)$$

$$\text{where } \mathcal{L}_* = [\beta \mathcal{L} - \alpha \mathbf{I}]^\rho \, .$$

Here, $x$ corresponds to an input RES, $y$ is an EES, and each of $\alpha, \beta$, and $\rho$ are parameters that control the spectral translation, reconstruction penalty, and filter order, respectively. MELD solves the problem of learning $y$ using a Chebyshev polynomial approximation. In contrast to previous works using Laplacian filters, these parameters allow analysis of signals that are contaminated by noise across the frequency spectrum, which is explored in Burkhardt et al. (2019).

Next, to identify populations of cells most or least affect by a given perturbation, we introduce a novel clustering algorithm called Vertex Frequency Clustering (VFC, **Fig. 1b**). VFC combines information about graph structure, RES signal frequency content, and EES signal magnitude at each node to identify clusters of cells that are both transcriptionally similar and affected similarly by the experimental perturbation. The implementation of VFC is described thoroughly in Burkhardt et al. (2019). At a high level, a series of varying-scale Windowed Graph Fourier Transforms are applied to the graph localized at each node to learn a spectrogram of the RES (Shuman et al., 2016). This matrix is concatenated with the EES to capture information about signal magnitude and used as input to k-means. As shown in Section 3.2, this approach can distinguish between intermediate cells between two extreme phenotypes and cells unaffected by an experimental perturbation (discussed in Section 3.2).

Once the EES and VFC clusters have been inferred, this information can be used to identify gene signatures on an experimental perturbation (**Fig. 1c**) and characterize cells types with varying response to perturbation (**Fig. 1d**). In the following section, we present application of both strategies on two single cell datasets.

## 3 RESULTS

### 3.1 MELD IDENTIFIES A BIOLOGICALLY RELEVANT SIGNATURE OF T CELL ACTIVATION

To demonstrate the ability of MELD to identify a biologically relevant EES, we applied the algorithm to 5740 Jurkat T cells cultured for 10 days with and without anti-CD3/anti-CD28 antibodies published by Datlinger et al. (2017). The goal of the experiment was to characterize the transcriptional signature of T cell Receptor (TCR) activation. We selected this data because it relatively simple: the experiment profiles a single cell type, yet exhibits a heterogeneous continuum of experimental responses. We visualized the data using PHATE, a visualization and dimensionality reduction tool we developed for single-cell RNA-seq data (**Fig. 2a**; Moon et al. (2018). We observed a large degree of overlap in cell states between the experimental and control conditions, as noted in the original study (Datlinger et al., 2017). This noise is both technical and biological. Approximately 76% of the cells were transfected with gRNAs targeting proteins in the TCR pathway, leading to some cells in the stimulated condition lacking key effectors of activation. The expectation for these cells is to appear transcriptionally unactivated despite originating from the stimulated experimental condition. In other words, although the RES for these cells is +1 (originating from the stimulated condition), the EES of these cells is expected to be closer to -1 (prototypical of the unstimulated condition).

To obtain a signature of T cell activation, Datlinger et al. (2017) devised an *ad hoc* iterative clustering approach whereby cells were first clustered by the gRNA observed in that cell and then further clustered by the gene targeted. In each cluster, the median gene expression was calculated and the first principle component was used as the dimension of activation. The 165 genes with the highest component loadings were defined as signature genes and used to judge the level of activation in each cell. We reasoned that MELD would be able to identify an EES of TCR activation without relying on clustering or access to information about the gRNA observed in each cell.

Applying MELD to the data, we observe a continuous spectrum of scores across the data set (**Fig. 2a**). As expected, the regions enriched for cells from the stimulated condition have higher

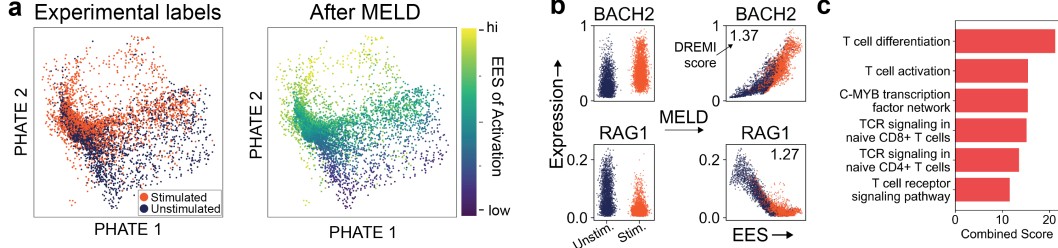

Figure 2: MELD recovers signature of TCR activation.

EES values representing highly activated cells, and the converse is true for regions enriched for unstimulated cells. To ensure that the EES represents a biologically relevant dimension of activation, we looked for genes with a high mutual information with the EES using kNN-DREMI (van Dijk et al., 2018). To facilitate comparison with the results of Datlinger et al. (2017), we used EnrichR (Kuleshov et al., 2016) to perform gene set enrichment analysis on the 165 genes with the top kNN-DREMI scores (**Fig. 2b**). We found comparable enrichment for gene sets related to T cell activation, T cell differentiation, and TCR response (**Fig. 2c**) and identify an overlap of 53 genes between the MELD-inferred and published signatures. We find that in the GO sets of T cell activation, T cell differentiation, and T cell receptor signalling, the MELD signatures includes as many or more genes for each GO term. Furthermore, our signature includes genes known to be affected by TCR stimulation but not present in the Datlinger et al. (2017) signature list, such as down regulation of RAG1 and RAG2 (Turka et al., 1991). These results show that MELD is capable of identifying a biologically relevant dimension of T cell activation at the resolution of single cells.

## 3.2 MELD AND VFC CAPTURE EFFECT OF IFNG STIMULATION IN PACREATIC ISLET CELLS

To demonstrate the ability for VFC to identify biological populations with various responses to perturbation, we analyzed a newly generated scRNA-seq experiment of human pancreatic islets stimulated with interferon-gamma (IFNg). Human islets from three donors were cultured for 24 hours with or without IFNg before collection for scRNA-seq. We visualized the data using PHATE and used MELD to infers a cell type specific response to treatment. Next, we used VFC to identify clusters of cells with similar response to IFNg. We calculated a spectrogram from the RES, concatenated the EES, and calculated k-means to obtain 9 VFC clusters (further description in Burkhardt et al. (2019)). Examining the resulting cluster, we identified alpha, beta, and delta cells (**Fig. 3a**). In each cell type, we observed

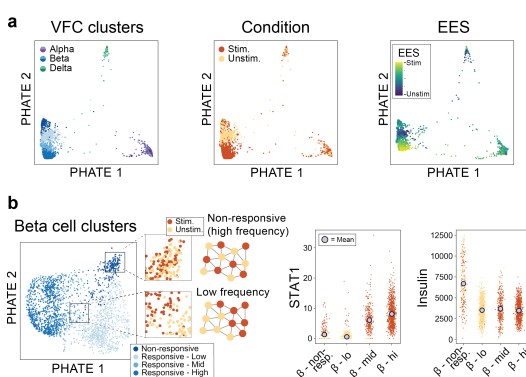

Figure 3: **(a)** VFC and MELD characterize response to IFNg across islet cell types. **(b)** Zooming in on beta cells, VFC identifies non responsive cells unaffected by stimulation.

that genes strongly correlated with the EES include known downstream targets of IFNg signalling including STAT1, IRF1, and ISG20 (data shown in Burkhardt et al. (2019)).

Zooming in on the beta cell clusters, we observed four groups of cells: cells enriched in either the stimulated or unstimulated condition, cells intermediate of these groups, and a group of cells with high-frequency RES content suggesting no response to IFNg (**Fig. 3b**). To confirm these clusters are biologically relevant, we examined expression of STAT1 and found that the cells enriched in the unstimulated condition and non-responsive cells have the lowest expression of this IFNg-induced gene. This indicates that the non-responsive cluster, despite containing roughly equal number of stimulated and unstimulated cells, exhibits no stimulation phenotype. We also find that the non-responsive cells are marked by extreme high insulin expression. Recent studies have described a subpopulation of beta cells marked by high insulin mRNA production that are hypothesized to have

functional differences to typical beta cells Farack et al. (2019). The results demonstrate the ability of MELD and VFC to tease apart subpopulations of cells exhibiting diverse responses to perturbation.

## 4 CONCLUSION

MELD introduces a novel and flexible filter on graph frequency domain to remove noise from the experimental labels indicating from which condition each cell was sampled. We show that this filter is capable of recovering unique signals from data with multiple structures. We also demonstrate the ability of MELD and VFC to identify biologically relevant signals across multiple cell types and biological systems.

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
