# OpenReview forum: "Enhancing experimental signals in single-cell RNA-sequencing data using graph signal processing"
_ICLR.cc/2019/Workshop/LLD — LLD 2019_

### Official Review · AnonReviewer2 · 2019-04-07
**Weak accept**

**Rating:** 3
**Confidence:** 1

**Review:**

* Content:

This paper introduces several methods to process experimental results on biological cells. These experiments are characterized by a high experimental variability, which makes difficult to process results. The proposed MELD algorithm maps hard group assignments (e.g. treatment/control in {-1, 1}) to soft assignments (in [-1, 1]) thanks to a low-pass filtering based on a graph built using data related to each cell. This later allows the authors to cluster relevant groups of cells, leading to biological insights.
Note that the paper is an excerpt of the bioRxiv paper of (Burkhardt et al.) (cited in the paper).

* Comment:

This paper is quite dense and makes a heavy use of technical acronyms, but it remains understandable and is well written besides that.

My main concern is related to the experimental validation of the method by the authors, which appears quite qualitative to me. Indeed, the authors mainly observe that the proposed method allows them to gain biological insights on some past experiments. While this is interesting from a biological perspective, from a machine learning perspective a more thorough benchmarking of MELD would have been appreciated. Maybe using a synthetic model would help understand its possible weaknesses?

Despite this concern, I would still vote in favor of acceptance for this paper, as methods removing "noise" from data to increase its "signal" ratio would probably be of interest to the LLD community.

---

### Official Review · AnonReviewer1 · 2019-04-08
**The paper is clear, the method is properly described and evaluated.**

**Rating:** 3
**Confidence:** 2

**Review:**

It is not clear how it is appropriate to the audience of the workshop because the data are properly labeled. Yet, the authors argue that there is noise in the data labeling.

1/ Introduction
Context and goals are clearly described.

2/ The MELD algorithm
Clear.
Then, they introduce Vertex Frequecy Clustering but the method is not detailed.

3/ Results
It seems good yet it's hard to judge because there is no comparison with other methods.
Authors used 'we' + citation which breaks double-blind review...

---

### Decision · Program_Chairs · 2019-04-16
**Acceptance Decision**

Accept